# Experiences and perspectives on rapid-test diagnosis of tuberculosis, histoplasmosis and cryptococcosis in people with advanced HIV/AIDS disease in Porto Alegre, Brazil

**Angelo Brandelli Costa**[1,2], **Laura dos Santos Boeira**[2], **Damião Soares de Almeida-Segundo**[3], **Lara Wiehe Chaves**[1], **Laura Sainz**[2], **Larissa Silva**[4], **Leonardo Mello Garcia Dos Santos**[5], **Nicole Reis**[4,6], **Alessandro C. Pasqualotto**[4,7], **Omar Sued**[8], **Freddy Perez**[4,8] *

1 Graduate Program in Medicine and Health Sciences, Pontifical Catholic University of Rio Grande do Sul, Porto Alegre, Rio Grande do Sul, Brazil, 2 Graduate Program in Psychology, Pontifical Catholic University of Rio Grande do Sul, Porto Alegre, Rio Grande do Sul, Brazil, 3 Graduate Program in Psychology, Federal University of Rio Grande do Sul, Porto Alegre, Rio Grande do Sul, Brazil, 4 Federal University of Health Sciences of Porto Alegre (UFCSPA), Porto Alegre, Rio Grande do Sul, Brazil, 5 Graduate Program in Sociology and Political Sciences, Pontifical Catholic University of Rio Grande do Sul, Porto Alegre, Rio Grande do Sul, Brazil, 6 Hospital Vila Nova, Porto Alegre, Rio Grande do Sul, Brazil, 7 Santa Casa de Porto Alegre, Porto Alegre, Rio Grande do Sul, Brazil, 8 Pan American Health Organization, Washington DC, United States of America

* perezf@paho.org

## Abstract

The rapid diagnosis of opportunistic infections (OIs) is critical for improving the health outcomes of people living with HIV/AIDS (PLWHA). This study aimed to describe the feasibility of implementing a package for the rapid diagnosis of tuberculosis, histoplasmosis, and cryptococcosis in patients with advanced HIV/AIDS disease in Porto Alegre, Brazil. The research involved two focus groups with health professionals, four in-depth interviews with healthcare managers, and twelve interviews with PLWHA. The corpus was analyzed using Descending Hierarchical Classification (DHC). The study found that the rapid test diagnosis intervention was generally well-received by patients and health professionals, improving diagnosis and treatment outcomes. However, it also identified several areas for improvement, including the need for expanded psychosocial support and enhanced coordination between health services. The findings have important implications for the development and implementation of policies and programs aimed at enhancing the diagnosis and treatment of OIs among PLWHA with advanced diseases. Further research should explore social determinants of HIV/AIDS mortality to offer valuable insights into improving prevention and treatment strategies. By prioritizing patient-centered care and improving coordination between health services, policymakers and health professionals can improve the health outcomes of PLWHA with advanced disease in Porto Alegre and other similar settings.

**Data Availability Statement:** Interview and focus group data cannot be fully disclosed due to the nature of the consent signed by participants, which did not include the provision for fully anonymized transcripts to be made available. Given the limited number of health professionals and public managers involved in the topics discussed in this manuscript in Porto Alegre, anonymization efforts would still not prevent the identification of participants if the full transcripts were released. This would risk violating the data privacy requirements set by the university research ethics committee (PUCRS). Data requests may be directed to the Pontifical Catholic University of Rio Grande do Sul University (Pontifícia Universidade Católica do Rio Grande do Sul) Ethics Committee at cep@pucrs.br.

**Funding:** This work was partially funded by the CDC-PAHO Agreement: Building Capacity and Networks to Address Emerging Infectious Diseases in the Americas [Award: 6 NU50CK000494-01-04; Funding Opportunity Announcement: CDC-RFA-CK18-1801CONT20] to FP. The funders had no role in study design, data collection and analysis, decision to publish, or preparation of the manuscript. This study is part of the memorandum of understanding work plan between the Pan-American Health Organization and the Federal University of Health Sciences of Porto Alegre (UFCSPA), Porto Alegre, Brazil.

**Competing interests:** The authors have declared that no competing interests exist.

## Introduction

Despite all advancements and innovations in HIV, as of 2022, 68% of individuals living with HIV still die from HIV-related AIDS, including opportunistic infections (OIs) globally. Leading causes of death include tuberculosis, histoplasmosis, and cryptococcosis [1]. In Brazil, the mortality rate is 4,1 per 100,000 inhabitants [2], but in the city of Porto Alegre it is five times higher (23,8 per 100,000 inhabitants).

Tuberculosis, histoplasmosis and cryptococcosis pose challenges to conventional diagnose methods like culture and histopathology [3–5]. Rapid diagnostic tests (RDTs), recommended by the World Health Organization (WHO) [6, 7], offer point of care (POC) solution by detecting antigens, which are associated with enhanced patient care and survival [5, 8–14]. However, test availability alone does not address barriers in care.

Qualitative health studies play a crucial role in comprehending the experiences and perceptions of key actors during the implementation of health innovations. These studies unveil valuable insights into various aspects, including the challenges faced by patients, the dynamics of interactions between health professionals and patients, the influence of social and cultural factors on health outcomes, the quality of health services, the effectiveness of interventions, and barriers hindering equitable access to care. In the evaluation of innovative technologies, qualitative approaches are instrumental in understanding factors that influence preferences, uptake, and acceptability among different user profiles, as well as capturing the experiences, and perspectives of providers and policymakers [15–17].

Since 2022 the "Cohort study for the rapid diagnosis of tuberculosis, histoplasmosis and cryptococcosis in people with advanced HIV disease" has been conducted at four tertiary care units in Porto Alegre, Southern Brazil. The project aims to evaluate the implementation of an expanded package of RDT-POC for detecting cryptococcosis, histoplasmosis, and TB among PLHIV with advanced HIV disease. Between January and December 2023, participants of this cohort underwent a testing intervention that included VISITECT CD4 Advanced Disease (Accubio), CrAg LFA (IMMY), TB LAM Ag (Abbott) and Histoplasma urine antigen LFA test (MiraVista) testing. Additionally, sputum and other materials were tested using the GeneXpert MTB/RIF (Cepheid). Patients were invited to participate in the testing by medical professionals of the tertiary care units. Laboratory professionals responsible for interpreting point-of-care test results received training for that purpose by a central team. Follow-up was conducted at 30 and 90 days to assess overall survival and the occurrence of severe adverse events.

The objective of this study is to provide insights, experiences, and perspectives on the feasibility of the proposed intervention among PLWHA. This study aims to contribute to the development and implementation of HIV/AIDS care policies, as well as to fulfill the overarching goal of enhancing the regional and national response to HIV/AIDS.

## Materials and methods

### Study population

**Patients.** A random subsample of patients from the cohort study was invited to participate based on inclusion criteria: being over 18-year-old, having HIV infection with signed informed consent, and meeting criteria for confirmed or suspected advanced HIV/AIDS disease (CD4 cells < 200 cells/mm3 in the last 3 months or any symptom suggestive of systemic infections in the last 14 days). Patients with ineffective ART (naive or less than three months of AR, treatment abandonment (>3 months) or virological failure (two consecutive detectable HIV viral loads, with at least one viral load being >1,000 copies/ml) were included. Individuals receiving active treatment for both tuberculosis and systemic fungal diseases in the past two

weeks were excluded from the study [18]. Patients from all four tertiary care units in Porto Alegre participating in the main study were invited to join the qualitative arm.

**Health professionals.**   Health professionals involved in the care of patients meeting the inclusion criteria were invited to participate. This group included doctors, nurses, medical students, and laboratory technicians. Professionals from all four tertiary care units in Porto Alegre participating in the main study were invited to join the qualitative arm.

**Policymakers.**   Policymakers, representing the HIV/AIDS Coordination at the Ministry of Health, the State Health Secretariat of Rio Grande do Sul and the Municipal Health Secretariat of Porto Alegre, as well as the coordinator of primary health care at the municipal level in Porto Alegre were invited. In case of unavailability, the invitation was extended to permanent employees with at least four years of experience in the technical area.

**Ethical procedures and patient involvement.**   Participation in the study was voluntary and required a signed informed consent. The study was based on a master protocol approved by the Pan-American Health Organization Ethical Committee (PAHOERC) under register PAHOERC.0347.01. A specific protocol for the qualitative arm was approved by the ethical committee of the Pontifícia Universidade Católica do Rio Grande do Sul (Approval number 6.177.483/2023). No personal identifiable information was collected in the database used for the analysis. Data was compiled by trained professionals and stored in a secure encrypted database on the university server designed for research data.

**Data acquisition.**   Participants completed a standard sociodemographic questionnaire. Semi-structured interviews were conducted with patients and health policymakers, and two focus groups were conducted with health professionals. Each session followed specific scripts and was recorded for transcription. Recruitment took place from 15 July 2023 to 15 August 2023. The scripts covered five central themes. Three themes were common to all stages, with each subsample emphasizing different aspects: rapid diagnostic interventions, experience with the intervention, and challenges/recommendations. Health professionals addressed the impact on clinical practice and interdisciplinary collaboration. Policymakers focused on the development of policy flows and inter-federative relationships. Patients discussed previous diagnostic experiences and perceived benefits of the intervention. The full protocol is available in S1 Appendix.

**Patient interviews.**   Initially, fifteen profiles of patients with diverse sociodemographic and clinical backgrounds were drawn up. However, due to difficulties in identifying these profiles using the registration database, 89 patients were randomly contacted, and 12 were interviewed. A list with anonymized codes representing each patient was provided by the hospitals, and a random draw was conducted by the research team to select 90 patients for contacted. Only after this selection did the team gain access to personal data such as names and telephone numbers. S2 Appendix contains characteristics of interviewed patients. Un two cases, relatives participated in the interviews due to the patients' inability to recall l events related to the intervention. One interview encountered technical audio issues, leading to notetaking during the conversation. The telephone interviews lasted approximately 40 minutes, and their transcripts comprised the study corpus.

**Focus groups with healthcare professionals.**   Two 120-minute focus groups involved 11 participants, including seven physicians, one nurse, two medical students and a laboratory technician. The corpus comprised transcripts of the professionals' conversations.

**Interviews with public policymakers.**   Four interviews were conducted with public HIV care policymakers at different management levels. Each interview lasted around 40 minutes, and their transcripts formed the corpus.

### Data analysis

Data cleaning included spell checks, standardizing word variations with the same meaning and the removal of linguistic mannerisms. Compound expressions that should be analyzed together were identified and merged. Additionally, terms indicated that people living with HIV were transformed into the acronym PLWHA.

In some cases, part of the interviewer's triggering question was retained in the corpus to anchor short answers from the patient, such as, yes or no responses to the question about a previous diagnosis of OIs. The corpus underwent analysis using Descending Hierarchical Classification (DHC), with IRAMUTEQ. DHC enables category creation based on word frequency, semantic affinity, context, and the level of relationship between words, measured by the Chi-Square test. The corpus was divided into Text Segments (TS), typically three lines long, determined by the software based on the corpus size and the context of the words. Throughout the text, the terminology TS, and Elementary Context Units (ECU) were used interchangeably since they represent the same concept in different software (IRAMUTEQ and Alceste, respectively),

## Results

### Focus groups with healthcare professionals

Fig 1 provides an overview of the corpus. The analysis incorporated 281 TS, retaining 70.82% of the total (199 ECU), organized into four clusters. The corpus was divided into two sub-corpora, one formed by clusters 1 and 3 and the other by clusters 2 and 4. The dendrogram displays the 10 words with the highest chi-squared value that significantly adhered to the cluster ($p < .001$). Additionally, a table summarizing the responses to questions was created to facilitate the visualization of the main themes discussed in the focus groups (S3 Appendix).

**Cluster 1, "Diagnostic Specificity**", encompasses statements related to concerns about test specificity and the potential of false positives causing delays in decision-making. **Cluster 3, "Early Diagnosis and Intervention**", shares related emerging themes and belongs to the same subcorpora as Cluster 1. Both clusters highlight the importance of rapid diagnosis for OIs, while expressing concerns about the impact of this on promptly initiating treatment for OIs. The other subcorpora focuses on similar themes around clusters 2 and 4. **Cluster 2, "Research and Team",** pertains to participants characterization, their fields of work and involvement in the intervention. **Cluster 4, "Intervention Potentials and Challenges**", addresses aspects related to the benefits and challenges of implementing new rapid tests.

**Cluster 1: Diagnostic specificity.**   This cluster comprises 27.64% of the ECU. The top ten words significantly associated ($p < .01$) with this cluster include: positive, result, come, negative, example, situation, need, take time, culture, and investigation. Cluster 1 revealed concerns about the diagnostic sensitivity and specificity of the tests. A prominent aspect within this theme is the sense of security when the diagnosis is positive and aligns with the team's suspicion. Conversely, there is difficulty and even distrust in cases where the disease is suspected, but the rapid test yields a negative result.

> I think the main thing we have seen in relation to the LAM test, okay? We often *see* a *positive result,* and, in fact, the *patient* has another non-tuberculous mycobacterium. We *saw this in* more than one *case, and we saw* patients with tuberculosis confirmed in more than one place (e.g., health center, hospital) and the *test came back negative*. [P4, FG1]

Providers acknowledged that OIs can mimic symptoms of other diseases, particularly histoplasmosis. They emphasized that the rapid test enables a quicker diagnosis, allowing patients

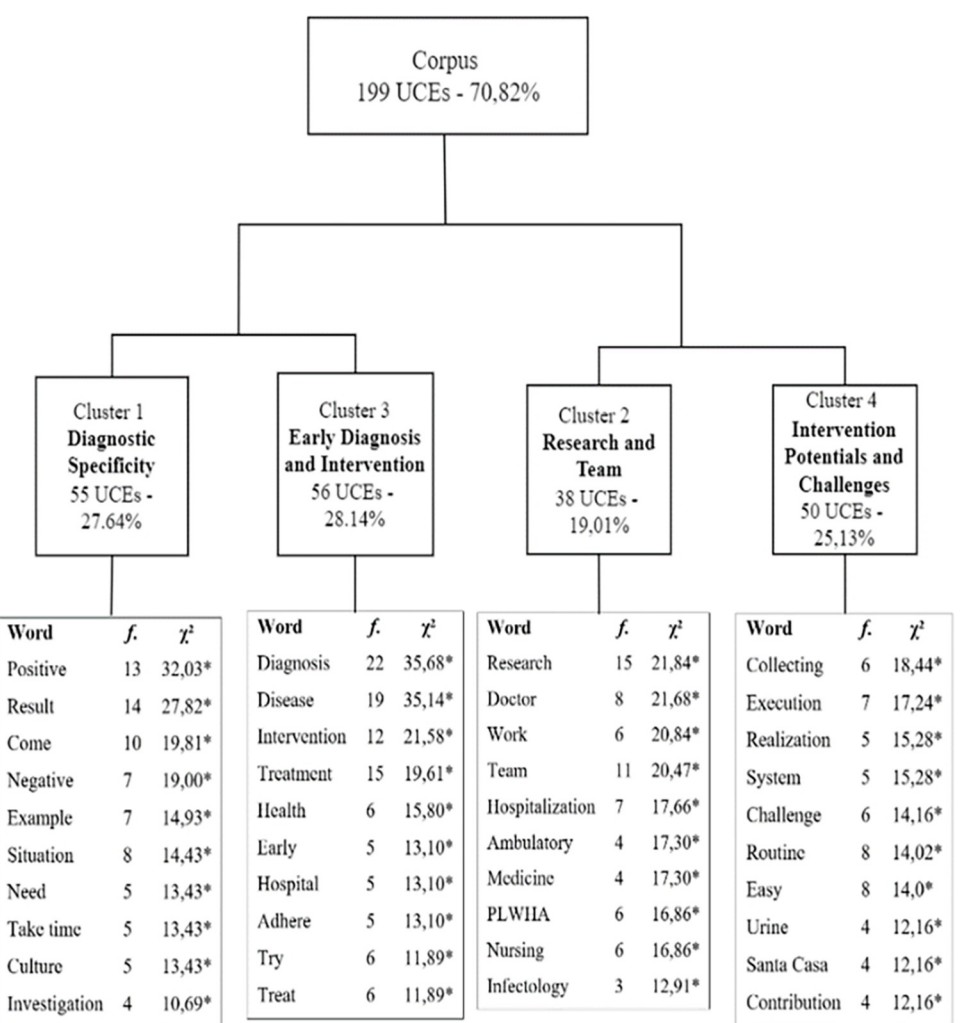

**Fig 1. Dendrogram of focus groups with health professionals.** Note. * p < 0.001.

to initiate treatment earlier and improve their prognosis. This perspective highlights the rapid diagnosis test as a complementary tool, providing confidence in the decision to start a potentially aggressive treatment in certain cases.

> [. . .] tuberculosis and cryptococcosis, we even had some therapeutic modalities that were faster, which does not *necessarily depend* on *culture*, but at least in our center, *histoplasmosis* changed a lot, and we *needed* tests that *would take time*, and we had a *clinical suspicion*, but it is still a disease that can mimic others. So, it was suspicion and treatment that added a lot of toxicity. For you to start simply from a suspicion. So sometimes it's good to have something. Some diagnostic suspicions and an antigen test that has good specificity and sensitivity with a positive result, confirming this hypothesis, gave us a lot of security. [P2, FG2]

**Cluster 3 –Early diagnosis and intervention.**    This cluster comprised 28.14% of ECU. The main terms significantly associated (p < .01) with the cluster were: diagnosis, disease, intervention, treatment, health, early, hospital, adhere, try, and treat. Similar to the TS that

adhered to Cluster 1, the content of Cluster 3 emphasizes the advantages of rapid diagnosis for OIs but primarily focuses on its impact on the quality of treatment rather than the sensitivity/specificity of the test. Some examples of the main speeches that joined this cluster are described below.

I think that the intervention *is* of significant importance because *it allows early diagnosis* of *diseases* potentially lethal *opportunist infections, whose diagnosis* is often difficult by traditional *means* [. . .]. [P2, FG1]

I said that *I would treat it* faster. We can minimize the negative consequences resulting from the *infection*. I stated that *rapid diagnostic intervention allows* for more targeted management in earlier stages of difficult-to-diagnose *diseases*. [P6, FG1]

These statements echo those in cluster 1, highlighting positive aspects and recognizing the significance of rapid diagnosis of OIs as a tool to facilitate quicker and more accurate diagnosis and treatment, ultimately leading to better prognosis. In Focus Group 2, Participant 3 presents a similar reflection, but with a more explicit concern for the other stages of treatment:

Well, it is not *time* that we always had available when *it came* to *treating* these patients, because they have a diagnosis of *serious illnesses*, with potential progression, to adverse outcomes, such as death. [. . .] If we don't institute *treatment* quickly, then I don't think the *way* of looking at advanced AIDS is changing much. When we enable *early diagnosis, initial treatment*, this changes the chain of *intervention and enables real* access and a possibility of cure. [P3, FG2]

**Cluster 2—Research and team.**   This cluster constituted 19.01% of ECU. The top words significantly associated (p < .01) were: research, doctor, work, team, hospitalization, ambulatory, medicine, PLWHA, nursing and infectiology. The content of this cluster is related to the team's experience in research, forming their impressions about the implementation. The main TS included challenges in coordinating clinical research between the regular staff and health professionals already working in the institution versus those specifically dedicated to research tasks (e.g. blood and urine collection). This led the hospital staff to take care of sone cases themselves.

[. . .] We had help from the nurses at first. And then, well, the problems started happening. The pee that was collected for the research wasn't sent to the laboratory by accident. It arrived at the laboratory, and the laboratory didn't understand why the pee had arrived at the laboratory. So, after that, we started to sort things out ourselves so that we were more in control of the flow. [. . .] And it is important to say that this was very *specific* to the *research*, the application of the tests, because if we are going to talk about *hospitalization* for advanced *AIDS*, we have gaps among the *multidisciplinary team* in all gaps and all spaces, from the pharmaceutic *to the nurse* [. . .]. [P3, FG2]

The provider emphasizes that the standard of care for PLWHA is a multidisciplinary team with a holistic approach. Another statement also highlights the tensions that appear in the care teams doing implementation research.

As for the difficulties [. . .]? It's actually being able to introduce this into something that isn't routine, right? So, we often end up annoying a lot of people. Like when we arrived and

asked for a test in the middle of the afternoon, because I need to take a test now. So, not everyone is very receptive when we arrive at this demand. With a certain. . . speed. [. . .] It turns out that you can't have the blood at one time and the urine at another. We do all the tests together. And if that's the case, P5 and I do it. We have to be at the hospital to do it, so that ends up being a. . . a difficulty, but in general, in most cases people are collaborative, and we manage to approach the issue in such a way that things go well, only one or two cases have been more complicated. [P6, FG1]

Also, the cluster identifies the importance of shifting tasks for routine implementation. For example:

I think that besides the *context* of the *research*, if the test was part of the routine, it would be much simpler because the *doctor* prescribes it, the *nurses apply it, by digital* puncture, it could be directly the test, urine collection, the result appears in the system on the same day. [P2, FG1]

**Cluster 4—Intervention potential and challenges.**  This cluster comprises 25.13% of ECU. The main terms significantly associated (p < 0.05) with the cluster were: collecting, execution, realization, system, challenge, routine, easy, urine and contribution. The content focuses on the potential for integrating standardized procedures to facilitate routine implementation and the challenges associated with implementing outside the research context. This includes the need for training, potential overload of human resources, ensuring financial resources, the manager's interest in implementing policies and assessing the cost-benefits of the intervention as a routine.

So, if this became *routine*, it would be *collected* in the three shifts of the *hospital*, so everyone, *people* would have to be trained, but also it is not *difficult* as it has not even been *implemented*, some tests are *easy to interpret* and *execute*. And the difficulties? I think it's being out of the routine because it interferes with the already established work demands of nursing and laboratory professionals. It's a question of costs and that's it [. . .]. [P1, FG1]

Finally, the concept of the importance on expanding access to health care for PLWHA with advanced HIV to these tests for making an impact: "I think the macro challenge here is to expand access to HIV care. Are we really managing to include all people who have advanced HIV? [. . .] What is happening? Who are these people? How does the test impact or not on this and based on that?". [P2, FG2]

## Interviews with policymakers

The DHC of these interviews is presented in Fig 2. This analysis included 417 TS, retaining 75.54% of the total (315 ECU), which were organized into three clusters. The corpus was divided into two subcorpora, one formed by clusters 1 and 3 and the other by Cluster 2.

**Cluster 1—Patient-professional relationship.**  This cluster was formed by 42.86% of ECU. The top words = significantly associated (p < 0.001) were: individual/subject, professional, point of view, testing, health, want, way, positive, result and realization. This cluster brought together discussions about the relationship between healthcare professionals and patients, specifically linked to rapid testing.

Manager 4 explains the importance of training human resources and expresses concerns about ensuring good patient care and maintaining the logic of the provider-user relationship:

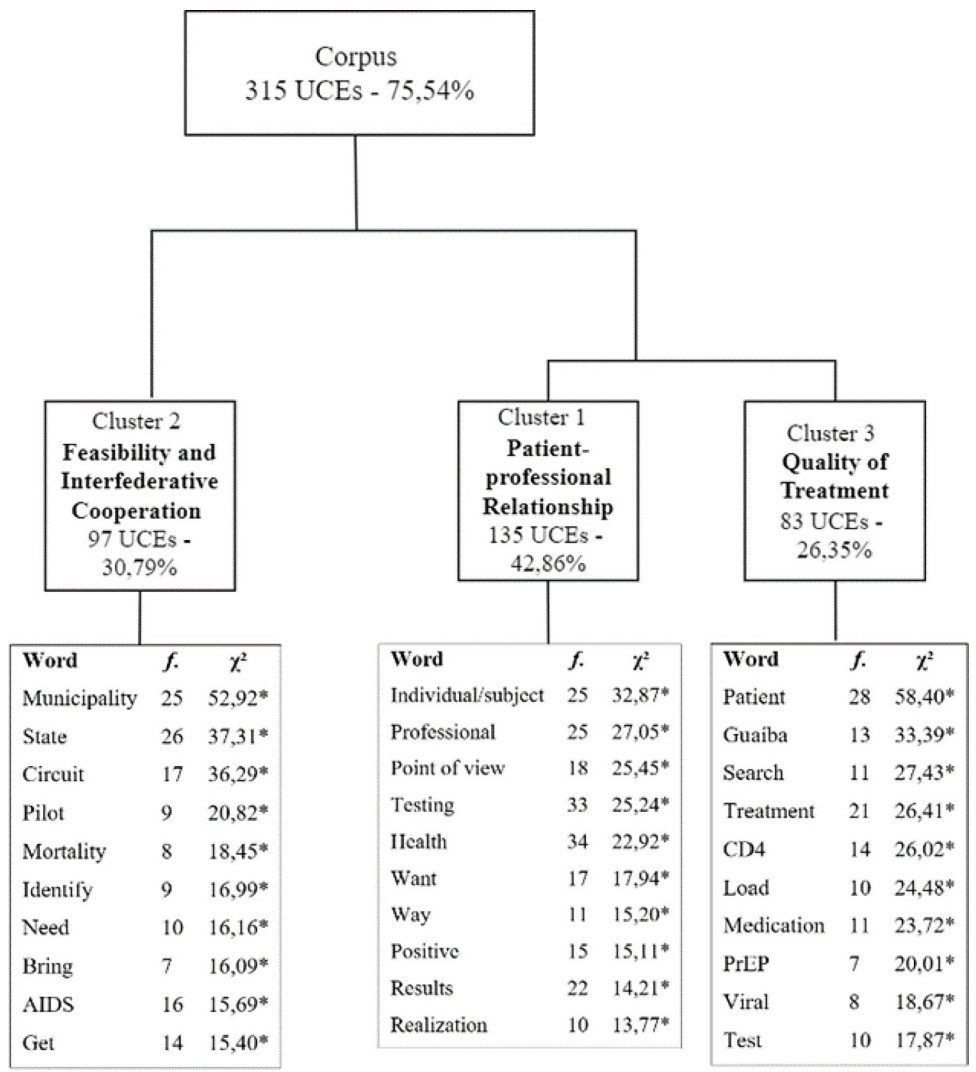

**Fig 2. Dendrogram of interviews with managers.** Note. * p < 0.001.

"But a health unit must be trained to provide testing, whether it's in the outpatient setting [. . .], or from the point of view of mobile testing, or testing at the health center [. . .]."

"But it *should not* be *fast* from the *point of view of user* care, patient care, respect for that patient. For us to have friendly environments to welcome *certain populations*, for us to welcome individuals who are looking for *rapid testing of OIs*."

"We still have a *logic* that the *health system* incorporates these *challenges* in *a* very automatic way, right? But this is not true. These *professionals* have their suffering, they have their own paradox in relation to the delivery of *tests*".

**Cluster 3—Quality of treatment.** This cluster focused on the importance of prompt treatment care quality, interventions for advanced HIV patients, and related challenges such as social vulnerability and substance abuse affecting adherence). It constitutes 26.35% of ECU. The main terms significantly associated (p < .001) to the cluster were: patient, search,

treatment, CD4, load, medication, PrEP (i.e., Pre-Exposure Prophylaxis), viral and test. Some aspects were linked to cluster 1. For instance, Manager 2 expressed concern about the low uptake of preventive TB treatment:

> "We really need to *expand LTBI (latent tuberculosis infection) test* coverage in the country as we realize that on a *daily basis*, those with a *CD4* lower *than* 350, who should receive *treatment*, don't do so and then the *patient starts* developing tuberculosis, right?".

Manager 1 empathizes with the importance of having systems to improve Antiretroviral Therapy (ART) adherence:

> "I collaborate directly with the pharmacist. In terms of *patients* who are abandoning treatment, which are the *patients* who are not *coming* to get *medication*? Why isn't he/she *coming*? So, we also plan some *things* like this to *make it easier* for them to adhere *to treatment*".

Finally, Manager 3 highlights the importance of the health networks within the health system, emphasizing the significant role of patient coordination and referrals:

> Once the rapid test is done, there must be a *CD4 viral load*, it turns out that this *CD4* is below 200. He needs to get to a specialized assistance service quickly, [. . .] So this entire network has to be very tightly knit, so we are going through specialized primary care, regulation, surveillance and the municipality's hospital board and coordination.

**Cluster 2—Feasibility and inter-federative cooperation.** This cluster addresses the challenge of integrating testing for OIs into the healthcare system including logistics, financial, and human resources considerations. It also explores inter-federative cooperation, involving collaborative efforts between different levels of government within a federal system (federal, state, and municipal governments). This cluster comprises 30.79% of the ECU. The top words significantly associated (p < .001) to Cluster 2 were: municipality, state, circuit, pilot, mortality, identify, need, bring, AIDS and get. Manager 3 expressed concern, stating:

> At this first *moment*, as we are working with a *pilot*, the *national policy* of the *Ministry of Health brings* to five *states* and some eligible *municipalities in the states*, the *pilot* of the *advanced* AIDS *circuit*. We do not have wide availability of supplies. For us, it is just another *technology* inserted within the *circuit* [. . .]

During the discussion, it became evident that the scarcity of human resources is a bottleneck for providing quality health services and could limit the benefits of expanding the provision of the tests, as expressed also by Manager 2

> "*Human resources always. States and municipalities and services always present human resources as one of the bottlenecks in not being able to get things done, right?*" . . . . . . "*So, I think it can be a strategy and the needs are infinite. Resources are finite.*"

In addition to resources, the network and inter-federative cooperation are critical according to Manager 3:

"So, this entire *network* must be very tightly knit, passing through specialized *primary care, regulation, surveillance and the city's hospital* management and coordination. There is no way to make a *circuit* of this magnitude without inter-federative collaboration.

## Patient interviews

Fig 3 displays DHC results with 394 TS, retaining 77.92% of the total (307 ECU), organized into four clusters. The corpus underwent three subdivisions to yield the final format with four clusters. Initially, the corpus split into two subcorpora, one containing Clusters 4 and the other of Clusters 3, 1 and 2 combined. This last subcorpora was further divided, with Cluster 3 on one side and Clusters 1 and 2 on the other. Finally, the sub-corpora involving Clusters 1 and 2 separated into distinct clusters. Additionally, a table was created to visualize the main interview themes (S4 Appendix).

**Cluster 4—Rapid diagnosis.** This cluster consolidates statements on the intervention, results, quality of life, communication adequacy, challenges, and improvement suggestions. The intervention received a positive evaluation, with concerns primarily focused on treatment rather than diagnosis. Key words (p < .001) included: rapid diagnosis, result, life, receive, information, expectation, sufficient, realize, follow up and method. Responses in this cluster were brief, and additional insights may be found in the summary table (refer to S4 Appendix). Excerpts from patients 9 and 3 include:

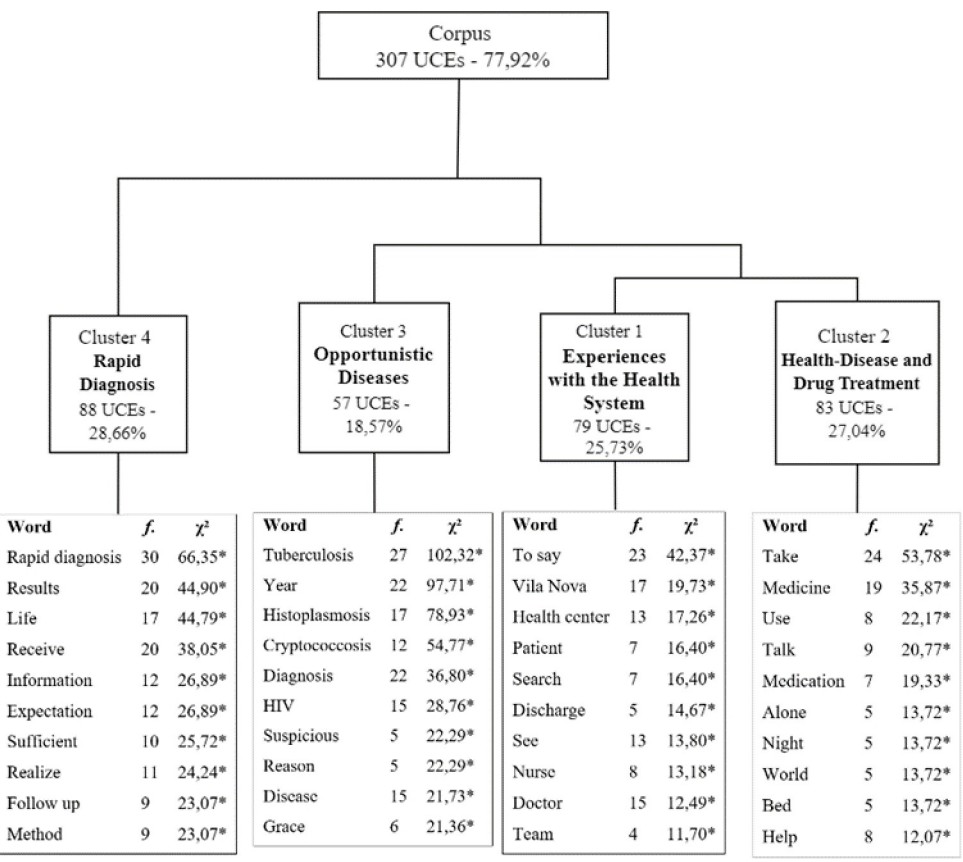

**Fig 3. Dendrogram from patient interviews.** * p < 0.001.

Interviewer: Did you face any difficulties in receiving the results of the rapid diagnosis?

Patient 9: No. It was all extremely fast.

Interviewer: Did you have any concerns about this intervention?

Patient 9: I had it with my family.

Interviewer: Did you face any difficulties in receiving the results of the rapid diagnostic?

Patient 3: No, I didn't find any difficulty with that.

**Cluster 3—Opportunistic infections.**   This cluster focused on topics related to diagnosis with participants' significant words (p < .001) including tuberculosis, year, histoplasmosis, cryptococcosis, diagnosis, HIV, suspicion, reason, disease, and grace. Like Cluster 4, this cluster provides limited information, with many responses being the simple yes/no. For example, after questioning about previous OI diagnosis, all patients reported with just "no," (refer to S4 Appendix) although patient 11 provided more detailed elaborations:

> My son had it first, my son was 17 years old at the time, he had it. And I had cancer, I had low immunity. I must have caught things like that, then. . . Having contracted tuberculosis and [. . .]. In the meantime, now, 5 months, 6 months ago, I went for the diagnosis and exam, and I found out that I was threatened with tuberculosis.

**Cluster 1—Experiences with the health system.**   Cluster 1's, predominate terms (p < .001) encompassed expressions like 'to say, 'health center", "patient", "search", "discharge", "see", "nurse", "doctor", and "team". These words capture experiences within the health system, particularly regarding rapid diagnosis and other health care procedures (e.g., exams, treatment). Patient 7 expressed concern about accessing continuous and quality care, highlighting system limitations that compelled multiple visits to the emergency room, describing the experience as moving: "from one place to another, like a headless chicken".

While patients expressed a lack of confidence and mistrust in non-infectious disease specialists, they acknowledged the impact of diagnostic tests on their treatment and appreciated the promptness of the results.

> Patient 4 "I didn't want a doctor who saw me at the hospital. He says I have diabetes, that I have high cholesterol. What!? I bought a device to test it, my blood glucose is seventy-five".

> Patient 8 "Then I went to hospital, the result of the test at the health center must have been a lie, then at hospital, he ran other tests there and saw that I was sick, and that's what happened?".

> Patient 6's "The doctor who treated me at the hospital was noticeably young. I was treated very, very well because I was able to go home with few consequences. I trust nurses because my son is a nurse."

> Patient 12's mother: "'Look, the service was very quick, it was correct, it was wonderful".

**Cluster 2—Health disease and drug treatment.**   Cluster 2 compiled statements regarding health-disease processes and drug treatment, including reports about illness due to OIs. The

most prevalent words in this cluster (p < .001) included "take", "medicine", "use", "talk", "medication", "alone", "night", "world", "bed", and "help". There were strong links with Cluster 1:

> Patient 10: I was treated for pneumonia. Antibiotics too, amoxicillin, I took other medicines they gave me, I took a lot of things, and it wasn't decreasing. It was important to go to the hospital, where they diagnosed tuberculosis.

> Patient 6's I had a psychotic break [. . .] I didn't know what it was. I was nervous [. . .] everything bad that happened to me in those days, I didn't remember anything[. . .]I take sleeping pills, and prescription drugs, I'm taking another one that's helping a lot. The pain is chronic.

## Discussion

Our study aimed to assess the feasibility of implementing a package for the rapid diagnosis of OIs in patients with advanced HIV in Porto Alegre, Brazil. The goal was to identify the challenges faced by patients and explore recommendations from health professionals and managers. Results suggest the intervention is feasible and effective as a strategy for improving outcomes among PLWHA with advanced disease. The study also highlights some important barriers and facilitators from the standpoint of our stakeholders.

Providers faced challenges with test specificity and expressed concern about false- positive results among asymptomatic individuals. Although false-positive results of rapid tests for opportunistic infections in advanced HIV are uncommon, they can occur due to technical or biological causes. Continuous education on interpreting test results, using clinician reference materials, and peer comparisons over time can enhance health professionals understanding of test specificity and minimize these concerns [19]. The WHO recommends a package of screening, prophylaxis, rapid ART initiation and intensified adherence interventions for everyone living with HIV presenting with advanced disease [20]. Providers emphasized the need for a quick test to expedite treatment, especially considering conventional tests for these diseases usually take many weeks. In this context, point of care, rapid tests appear as a promising tool. "The additional cost of performing the three antigen tests for all individuals with advanced HIV disease is minimal, as these tests do not require specialized infrastructure or laboratory capacities. Furthermore, for each test, there is published evidence supporting its cost-effectiveness [21–23].

Another challenge mentioned was integrating the intervention into the routine health system results in high work demand. Providers highlighted the need for training in simple and didactic methods about diagnosis and treatment, emphasizing interdisciplinary/multidisciplinary collaboration and task shifting for effective delivery. Peer-delivered communication was suggested as a potential alternative [24]. Policymakers echoed these concerns, emphasizing the importance of mandatory training in communication, diagnosis, and effective treatments to ensure the quality of care. When caring for advanced HIV patients, there's a crucial emphasis on the speed of diagnosis and initiation of treatments to significantly reduce early mortality.

Addressing human resources and forming a robust care network are crucial for the successful implementation of testing for OIs at both primary and secondary care levels. This includes the strategic deployment of specialized professionals such as infectologists, to meet patient demands. Regulating the health system to support these efforts requires establishing clear surveillance and management protocols to monitor the efficacy and reach of the testing package,

as well as developing comprehensive care and treatment protocols. Additionally, fostering inter-federative collaboration is essential to harmonize efforts across different regions and government levels, ensuring consistent implementation and access to testing services. By integrating these components, the health system can effectively respond to the challenges of testing for HIV and OIs and treatment, thereby improving patient outcomes and public health.

This study also revealed the significant impact of social and economic vulnerability on the health of certain patients, emphasizing the need to strengthen strategies to prevent abandonment and facilitate adherence. Suggestions included sending medication and providing transportation to re-engage individuals in ART. Providers and policymakers proposed that coordinating specialized primary care, regulation, surveillance, management and hospital efforts within the municipality, along with collaboration with systems such as the Brazilian Public System of Social Assistance (SUAS) is critical. Conditional cash transfer programs have demonstrated effectiveness reducing HIV-related mortality in Brazil [25].

Patients expressed a positive perception of receiving tests to expedite treatments, but challenges emerged, including health issues associated with advanced HIV (seizures, starvation, not being able to walk, lung pain), and the importance of receiving respectful care. Psychological aspects of relief were also noted, with phrases like "I'm hearing something good", "getting better", "knowing is good" and "resolving the worry". Patients suggested improvements including a) having specialized professional (infectologists) leading the intervention; b) availability of other tests and medicines at point-of-care with decentralization; c) availability to home care; d) availability of psychologists to help cope with the impacts of living with HIV; and e) better coordinated care.

This paper has some limitations. First, due to the nature of the research and low response to patient's telephone calls, participant diversity was limited. The number of providers from non-physician disciplines was also limited, with reasons for non-participation including time constraints, privacy concerns, and lack of motivation to contribute to the study. Phone recruitment and web-based interviews may have restricted access for those without service access. However, this method was chosen due to time constraints and coordination challenges, given the city's size and its metropolitan area, and potential difficulties in interviewing untreated and unwell patients with advanced HIV disease. Some strengths of the study include its integration within a large implementation trial, well-trained teams and standardized procedures, and the inclusion of a significant number of vulnerable patients from the site.

Based on our study findings, several recommendations can be made for the sustainability and improvement of the rapid diagnostic interventions for OIs in patients with advanced HIV/AIDS in Porto Alegre. At the local level, policymakers and health professionals should prioritize implementing the rapid-test diagnosis intervention for tuberculosis, histoplasmosis, and cryptococcosis in patients with advanced HIV/AIDS disease. This intervention should be complemented by clear communication, psychosocial support, and patient-centered care to ensure optimal patient outcomes. Additionally, policymakers should focus om enhancing coordination between health services to guarantee prompt and proper care for patients. Addressing the social determinants of HIV/AIDS mortality, including racial, educational and income inequities, to crucial for ensuring that vulnerable key-populations have access to the care and support needed for healthy and fulfilling lives [26–28].

There are several key strategies to improve stakeholders' adherence to the intervention. First, increasing task shifting can reduce the burden on HIV providers by delegating certain responsibilities to trained non-specialist health workers, thereby improving efficiency and access [29]. Engaging hospital managers to fully understand the benefits of the intervention is essential, as their support can drive institutional adoption and resource allocation. Integrating these testing interventions into emergency room protocols ensures that more patients are

reached in critical settings. Simplifying the testing circuits and reporting processes will stream-line operations, making it easier for healthcare providers to comply with protocols. Finally, ongoing discussions with federal and local governments are crucial to secure the continued provision of tests, ensuring the program's sustainability and broad accessibility.

In conclusion, this article offers a comprehensive analysis of the feasibility of implementing a package for the rapid diagnosis of OIs in patients with advanced HIV in Porto Alegre, Brazil. Through interviews with health professionals, managers, and patients, the study identified the primary challenges faced by patients in obtaining rapid diagnosis, as well as the recommendations of health professionals and managers for the implementation of HIV care policies in the region.

## Supporting information

**S1 Appendix. Semi-structured interview/focus group script.**
(DOCX)

**S2 Appendix. General characteristics of patients.**
(DOCX)

**S3 Appendix. Summary table of professional groups.**
(DOCX)

**S4 Appendix. Summary table of interviews with patients.**
(DOCX)

## Acknowledgments

The authors thank all patients, health care workers and health managers for their availability to participate in this study.

## Author Contributions

**Conceptualization:** Angelo Brandelli Costa, Laura dos Santos Boeira, Omar Sued, Freddy Perez.

**Data curation:** Angelo Brandelli Costa, Laura dos Santos Boeira, Damião Soares de Almeida-Segundo, Larissa Silva, Nicole Reis, Alessandro C. Pasqualotto, Omar Sued.

**Formal analysis:** Angelo Brandelli Costa, Laura dos Santos Boeira, Damião Soares de Almeida-Segundo.

**Funding acquisition:** Omar Sued, Freddy Perez.

**Investigation:** Laura dos Santos Boeira, Lara Wiehe Chaves, Laura Sainz, Leonardo Mello Garcia Dos Santos.

**Methodology:** Angelo Brandelli Costa, Laura dos Santos Boeira.

**Project administration:** Laura dos Santos Boeira, Larissa Silva, Nicole Reis, Alessandro C. Pasqualotto, Omar Sued, Freddy Perez.

**Resources:** Omar Sued, Freddy Perez.

**Supervision:** Angelo Brandelli Costa, Laura dos Santos Boeira, Larissa Silva, Nicole Reis, Alessandro C. Pasqualotto, Omar Sued, Freddy Perez.

**Validation:** Angelo Brandelli Costa, Laura dos Santos Boeira, Damião Soares de Almeida-Segundo, Omar Sued, Freddy Perez.

**Visualization:** Damião Soares de Almeida-Segundo.

**Writing – original draft:** Angelo Brandelli Costa, Laura dos Santos Boeira, Damião Soares de Almeida-Segundo, Lara Wiehe Chaves, Laura Sainz, Leonardo Mello Garcia Dos Santos.

**Writing – review & editing:** Angelo Brandelli Costa, Laura dos Santos Boeira, Damião Soares de Almeida-Segundo, Lara Wiehe Chaves, Laura Sainz, Larissa Silva, Leonardo Mello Garcia Dos Santos, Omar Sued, Freddy Perez.

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
