## [Decision Letter · Decision Letter 0]

19 Aug 2024

PONE-D-24-25548Experiences and perspectives on rapid-test diagnosis of tuberculosis, histoplasmosis and cryptococcosis in people with advanced HIV/AIDS disease in Porto Alegre, BrazilPLOS ONE

Dear Dr. Perez,

Thank you for submitting your manuscript to PLOS ONE. After careful consideration, we feel that it has merit but does not fully meet PLOS ONE’s publication criteria as it currently stands. Therefore, we invite you to submit a revised version of the manuscript that addresses the points raised during the review process.

The study presents a methodology appropriate to the objectives and the manuscript is well written, although revision is necessary, as in line 462: a) ....... (infect ologists). Its results provide a contribution to the implementation of a screening and treatment strategy recommended by the WHO in a Brazilian metropolis, however, some aspects could be better explored in the discussion as healthcare workers: training of health professionals and multidisciplinarity of the health team; as policemakers: necessary human resources and formation of a care network (primary, secondary), regulation of the health system, surveillance and management of the system, including the demand of patients for specialized professionals (infectologists) and formulation of care and treatment protocols, as well as interfederative collaboration.

Please clarify how the random selection of participants (patients) for the interview was carried out?

We look forward to receiving your revised manuscript.

Kind regards,

Maria Carlota Borba Brum, PhD

Academic Editor

PLOS ONE

4. In the online submission form, you indicated that [The data underlying the results presented in the study are available from request of the correspondent author.]. 

Additional Editor Comments:

The study presents a methodology appropriate to the objectives and the manuscript is well written, although revision is necessary, as in line 462: a) ....... (infec tologists). Its results provide a contribution to the implementation of a screening and treatment strategy recommended by the WHO in a Brazilian metropolis, however, some aspects could be better explored in the discussion as healthcare workers: training of health professionals and multidisciplinarity of the health team; as policemakers: necessary human resources and formation of a care network (primary, secondary), regulation of the health system, surveillance and management of the system, including the demand of patients for specialized professionals (infectologists) and formulation of care and treatment protocols, as well as interfederative collaboration.

Please clarify how the random selection of participants (patients) for the interview was carried out?

Reviewers' comments:

Reviewer's Responses to Questions

**Comments to the Author**

1. Is the manuscript technically sound, and do the data support the conclusions?

Reviewer #1: Partly

Reviewer #2: Yes

2. Has the statistical analysis been performed appropriately and rigorously? 

Reviewer #1: I Don't Know

Reviewer #2: Yes

3. Have the authors made all data underlying the findings in their manuscript fully available?

Reviewer #1: Yes

Reviewer #2: Yes

4. Is the manuscript presented in an intelligible fashion and written in standard English?

Reviewer #1: Yes

Reviewer #2: Yes

5. Review Comments to the Author

Reviewer #1: Clearly describe what type of secure software was used to store the survey data.

Describe the cost of implementing the diagnostic package.

Present and describe proposals to intensify adherence to the proposed intervention based on the difficulties highlighted in the study.

Reviewer #2: The study was well conducted from a scientific standpoint, and the manuscript is very well written. Its results provide a significant contribution to understanding how the implementation of a 'screen and treat' strategy recommended by the WHO was perceived by patients, healthcare professionals, and managers in a Brazilian metropolis. This knowledge, which goes beyond the accuracy of the tests used, offers crucial scientific support for improving the implementation of this strategy, considered a priority in reducing AIDS-related deaths worldwide.

I have three suggestions to improve the manuscript:

1. In the Methods section:

a. To help the reader better understand the perceptions of patients, healthcare professionals, and managers regarding the strategy of screening for opportunistic infections with rapid tests (rapid diagnostic intervention), the authors should provide a detailed description of how this strategy was carried out. This includes which tests were used, how long it took, how the tests were conducted, and by whom.

b. The authors should clarify in the methods section from which settings the patients and healthcare professionals were selected, just as they did when describing the selection of the managers.

2. In the Discussion section: Healthcare professionals expressed concerns about the accuracy of the tests, which could negatively impact the implementation of the strategy. I suggest that the authors discuss possible measures to minimize this issue in the paragraph on line 429.

6. PLOS authors have the option to publish the peer review history of their article (what does this mean?). If published, this will include your full peer review and any attached files.

Reviewer #1: **Yes: **Sheila de Castro Cardoso Toniasso

Reviewer #2: **Yes: **Anamaria Mello Miranda Paniago

---

## [Author Response · Author response to Decision Letter 0]

13 Sep 2024

All comments from the editor and reviewers have been covered as detailed in the document' Response to Reviewers' and the manuscript with tack changes.

---

## [Editor Report · Decision Letter 1]

23 Sep 2024

Experiences and perspectives on rapid-test diagnosis of tuberculosis, histoplasmosis and cryptococcosis in people with advanced HIV/AIDS disease in Porto Alegre, Brazil

PONE-D-24-25548R1

Dear Dr. Perez,

We’re pleased to inform you that your manuscript has been judged scientifically suitable for publication and will be formally accepted for publication once it meets all outstanding technical requirements.

Kind regards,

Maria Carlota Borba Brum, PhD

Academic Editor

PLOS ONE

Additional Editor Comments (optional):

Manuscript presents improvements, especially in the methods and discussion section, with relevant contributions on the implementation of the rapid testing program in the context of the territory.
---

## [Editor Report · Acceptance letter]

17 Oct 2024

PONE-D-24-25548R1 

PLOS ONE

Dear Dr. Perez, 

I'm pleased to inform you that your manuscript has been deemed suitable for publication in PLOS ONE. Congratulations! Your manuscript is now being handed over to our production team.

Kind regards, 

on behalf of

Dr. Maria Carlota Borba Brum 

Academic Editor

PLOS ONE